# Keeping Safe on Australian Roads: Overview of Key Determinants of Risky Driving, Passenger Injury, and Fatalities for Indigenous Populations

**DOI:** 10.3390/ijerph18052446

**Published:** 2021-03-02

**Authors:** Kristen Pammer, Melissa Freire, Cassandra Gauld, Nathan Towney

**Affiliations:** 1The School of Psychology, Faculty of Science, The University of Newcastle, Callaghan, NSW 2308, Australia; Melissa.Freire@newcastle.edu.au (M.F.); Cass.Gauld@newcastle.edu.au (C.G.); 2Vice-Chancellor’s Division, The University of Newcastle, Callaghan, NSW 2308, Australia; Nathan.Towney@newcastle.edu.au

**Keywords:** Australian Indigenous, road safety, passenger safety, seatbelt use, overcrowding

## Abstract

Social and cultural barriers associated with inequitable access to driver licensing and associated road safety education, as well as socioeconomic issues that preclude ongoing vehicle maintenance and registration, result in unsafe in-car behaviours such as passenger overcrowding. This in turn is associated with improper seatbelt usage, noncompliance with child restraint mandates, and driver distraction. For example, in Australia, where seatbelt use is mandatory, Indigenous road users are three times less likely to wear seatbelts than non-Indigenous road users. This is associated with a disproportionately high fatality rate for Indigenous drivers and passengers; 21% of Indigenous motor-vehicle occupants killed on Australian roads were not wearing a seatbelt at the time of impact. In addition, inequitable access to driver licensing instruction due to financial and cultural barriers results in Indigenous learner drivers having limited access to qualified mentors and instructors. A consequent lack of road safety instruction results in a normalising of risky driving behaviours, perpetuated through successive generations of drivers. Moreover, culturally biased driver instruction manuals, which are contextualised within an English written-language learning framework, fail to accommodate the learning needs of Indigenous peoples who may encounter difficulties with English literacy. This results in difficulty understanding the fundamental road rules, which in turn makes it difficult for young drivers to develop and sustain safe in-car behaviours. This paper considers the literature regarding road safety for Indigenous road users and critically evaluates strategies and policies that have been advanced to protect Indigenous drivers. Novel solutions to increasing road safety rule compliance are proposed, particularly in relation to passenger safety, which are uniquely embedded within Indigenous ways of knowing, being, and doing. Safe driving practices have crucial health and social implications for Indigenous communities by allowing more Indigenous people to participate in work and education opportunities, access healthcare, maintain cultural commitments, and engage with families and friends, qualities which are essential for ongoing health and wellbeing.

## 1. Introduction

The likelihood of being involved in a traffic accident is up to six times higher for Australian Indigenous peoples than for non-Indigenous Australians; Aboriginal and Torres Strait Islander peoples are 1.4 times more likely to suffer serious injury, and 2.7 times more likely to die as the result of a vehicle crash [1,2]. Furthermore, Indigenous passenger occupants are three times more likely to be fatally injured than non-Indigenous passenger occupants; passenger occupant fatalities account for over 61% of Indigenous transport fatalities [1]. This is despite Aboriginal and Torres Strait Islander peoples constituting little more than 3% of the Australian population [3]. Numerous statistical examinations of the risk factors in fatal and nonfatal car crashes identify alcohol use, non-use of seatbelt restraints, and passenger overcrowding as the top risk factors in death and serious injury [4]. While drink-driving has received considerable attention [5], very few studies have focused on the specific issues of wearing car restraints or overloading the vehicle. Of the Indigenous road users hospitalised as a result of a car crash, over one-quarter were passenger occupants, with the vast majority of injuries associated with failing to wear a seatbelt restraint. Some reports suggest that as many as 60% of fatalities were a consequence of being ejected from a vehicle from not wearing a seat restraint [6].

The majority of published studies and reviews in this area are specific to particular regions within Australia, such as the Pilbara region in remote Western Australia [7], remote Queensland [8], and Northern Queensland [9]. This undoubtedly reflects the unique pressures faced by remote Indigenous communities and highlights that a one-size-fits-all approach to intervention across Australia is unlikely to be successful. Similarly, the vast majority of work in this space is in the form of government reports rather than independent research. These focused research and reporting approaches, however, reduce the scope and generalisability of outcomes and recommendations. Thus, the aim here was to provide a more general overview of the emerging themes from all sources, in order to provide a broader understanding and basis for a more functional roadmap of potential interventions moving forward.

Systemic barriers limit access to road safety education and resources, which in turn contribute to an increased prevalence of risky driving behaviour and increase crash risk for Indigenous Australians. Risky driving behaviour is defined within the driving and road safety literature as driving behaviour that increases the potential for harm or serious injury (e.g., [10,11,12]). In many Indigenous communities, risky driving behaviour is fundamentally associated with broader systemic issues of socioeconomic disadvantage, cultural norms, and expectations, and it is often influenced by geographical remoteness. Behaviours that put passengers at risk are not the consequence of unique causes, but are tightly interwoven and complex; for example, failing to wear personal protective equipment such as seatbelts is not distinguishable from the issue of overcrowding cars. Rather, one is often a consequence of the other and—in this example—may also be influenced by Indigenous cultural expectations regarding reciprocity and obligation.

Consistent with the theme of this Special Edition, the primary focus of this review paper was understanding the risky driving behaviours that impact passenger safety for Indigenous Australians. Specifically, this paper focuses on the use of seatbelts, child restraints, and overcrowded cars, given that these behaviours are known contributing factors to death and serious injury among Indigenous peoples [3]. Importantly, we recognise that to clearly understand the determinants of these risky driving behaviours for Australian Indigenous populations, our review needs to be contextualised within the broader context of cultural, societal, and systemic issues that are unique to Australian Indigenous populations. As such, this paper considers the general context of risky driving practices in Australian Indigenous communities from a road-safety, sociocultural, and population perspective.

## 2. The Issue

Indigenous Australians represent approximately 3.3% of the total Australian population (approximately 798,400 people). This figure is projected to reach over 1.1 million people by 2031 (Australian Bureau of Statistics, 2016 reference period) who identify as being of Aboriginal (91%) or Torres Strait Islander origin (5%) or both (4%). The median age of Australian Indigenous peoples is 23 years, compared with 38 years for non-Indigenous Australians, indicating a population that is skewed toward younger people, with less than 4% of the Indigenous population over 65 years [13]. Transport-related injury is the second leading cause of death and the fourth leading cause of serious injury for Indigenous Australians [1]. Compared with non-Indigenous people, Indigenous Australians are 2.7 times more likely to be killed and 1.4 times more likely to be injured in a land transport crash [1]. Moreover, driver’s licence offences equate to the highest offence category for convictions of Indigenous peoples [14,15]. Figures on population incarceration suggest that more than 90% of people incarcerated for driving offences around Australia are Indigenous [16]. Similarly, a higher proportion of Indigenous passengers are killed and hospitalised compared to non-Indigenous passengers [17,18]. However, these statistics are likely underestimated due to some Australian states not requiring Aboriginal and Torres Strait Islander peoples to report cultural heritage when obtaining a driver’s licence or in crash data.

Risky driving behaviour resulting in a vehicular crash is, therefore, a major cause of death and serious injury for Indigenous populations in Australia, and repeated offences associated with risky driving behaviour represent one of the leading reasons for Indigenous incarceration. Moreover, this overrepresentation of Indigenous Australians in road crash and road trauma is consistent with First Nations peoples elsewhere around the world, such as New Zealand, United States, Canada, and South Africa [19,20,21,22]. Indigenous people’s commitment to family, place and land, reciprocity, and family responsibility are a fundamental basis of the requirement for travel, in that it is often a social imperative for many Indigenous community members to travel to connect with and care for extended family, share vehicles, and provide transport favours. The need to adhere to these social and familial obligations, coupled with individuals having limited access to vehicles, can provide impetus for the engagement in unsafe driving practices, a relationship that is illustrated in this paper. Furthermore, geographical isolation is a factor for many remote communities as it is associated with greater travel distances (in some cases, many thousands of kilometres between towns, communities, and homelands), usually on poorly maintained, unsealed roads and tracks. Indeed, given that many Indigenous peoples in remote and very remote areas travel on ungazetted roads, and crashes on ungazetted roads are not reported, available crash data relating to Indigenous peoples are likely to be underreported [4].

## 3. Systemic Barriers to Safe Driving in Indigenous Communities

Incidents of yearly reported driving offences across the Northern Territory of Australia [23] indicate that more than half are regulatory driving offences such as drink-driving, speeding, parking offences, failure to wear seatbelts, and failure to adhere to road rules. Vehicle registration and roadworthiness offences, as well as driving licence offences, constitute almost equal proportions of the remaining incidences. Distraction is an increasingly common risky driving behaviour, particularly the use of mobile phones. In a qualitative study of Indigenous crash victims in rural Northern Queensland, Edmonston [9] reported that in-car distraction was reported by crash victims in 35.4% of cases. However, alcohol intoxication remains the predominant risky driving factor in Indigenous road fatalities, with approximately 66% of such fatalities associated with a blood alcohol concentration (BAC) above the legal Australian limit of 0.05 [24].

While the focus of this paper is on understanding how risky driving behaviours of Indigenous peoples relate to passenger safety, we recognise and acknowledge that many of the risky in-car and driving behaviours of Indigenous community members, particularly those living in remote and very remote regions, are associated with ongoing systemic barriers that prevent equitable access to road safety education and services. Therefore, prior to discussing risky driving behaviours from a theoretical standpoint, it is necessary to discuss these systemic barriers to contextualise risky driving behaviours within a broader situation of inequity for Australian Indigenous peoples that is associated with intergenerational effects of colonisation in Australia. It is by understanding this broader context of inequitable access that we can come to understand that for Indigenous populations, unlike for other populations, engaging in risky driving behaviours often occurs out of necessity and does not necessarily stem from malevolent intent.

This inequitable access begins when Indigenous pre-drivers seek to obtain a driver’s licence; Australia, like many countries, requires pre-drivers to learn road rules and associated road safety information by studying a written road users’ manual, before demonstrating their understanding of the information by successfully completing a written test. This can be difficult for many Indigenous Australians who may not possess the literacy skills necessary to read and comprehend written material. In remote and very remote Indigenous communities, where English is typically a second or subsequent language for most community members, 76% of Indigenous children cannot read at the minimum national standard for their age [25]. Although there are little available data on literacy skills of Indigenous adults and no current reliable measure of Australian Indigenous adult literacy, one study in New South Wales (NSW) reported that 68% of 1177 surveyed Indigenous adults rated themselves as having low or very low English literacy skills, which indicates that, for many Indigenous people, English literacy barriers persist into adulthood [26]. This in turn negatively impacts an individual’s ability to understand road rules, road safety information, and the associated consequences of unsafe driving [17]. Intergenerational barriers to accessing road safety information due to literacy issues means that many older community members are unlicensed, which makes it difficult for young learner drivers to access driving mentors to gain the necessary driving experience required to successfully pass their on-road driver’s licensing test. This perpetuates a cycle that leads to many Indigenous drivers remaining unlicensed.

The geographical remoteness of many Indigenous communities hinders equitable access to employment, thus making it difficult for community members to acquire the financial means necessary to maintain safe, roadworthy, and registered vehicles. For example, a lack of financial resources, compounded by geographical isolation, makes it difficult to source appropriate child restraints. The cost of purchase and installation of an appropriate restraint is often prohibitive for many community members, particularly given that purchasing such items in remote communities can cost up to three times more than if purchased within a major city. In suburban areas, appropriate child restraints can be hired, but this service is not generally available in remote and rural regions of Australia. In addition, baby capsules must be installed and correctly fitted to anchorage points in a vehicle. Fitting can be conducted for free by qualified fitters; however, such resources are limited or unavailable in most regional and remote areas of Australia.

Despite barriers to obtaining driver’s licences and maintaining safe, roadworthy, and registered vehicles, access to safe and affordable private transport is necessary for many Indigenous populations in the context of sociocultural connectedness and kinship obligation. For example, the need to attend funerals and be with extended family members during “sorry business” can require people to travel to and between communities or homelands [27], as does the requirement to return to and care for traditional lands. This is particularly highlighted when individuals require medical treatment. Indigenous peoples have a greater prevalence of chronic disease, and remote communities cannot provide the services needed; therefore, individuals need to travel to larger regional areas. This can result in the need to drive vast distances, often over poorly maintained, unsealed dirt roads in a societal context where few communities have access to safe vehicles. It can also result in passenger overcrowding when many people need to travel to satisfy cultural and community obligations, but there are a limited number of available vehicles in the community.

Intergenerational and social norms play an important part in the perpetuation of unsafe driving behaviour, where intergenerational acceptance of more risky practices results in higher levels of socially sanctioned risky driving. This provides a psychological “framework” for Indigenous drivers and passengers to make “cost–benefit” judgements regarding the likelihood of crashing or being caught by the authorities. This is particularly the case for failing to wear seatbelts and overcrowding cars.

“*It wouldn’t have been so bad if they weren’t piled in the back of the tray but that’s what happens out here. You all pile in. It’s no big deal—everybody does it and the cops don’t worry*.”

“*… We were just unlucky—this is part of our ritual and usually nothing goes wrong*.”[9] (p. 10)

These aforementioned comments illustrate a social context where risky driving behaviour is accepted as the norm, resulting in a situation where Indigenous community members are less likely to engage in safe driving practices and may engage in risky driving behaviours with a sense of invulnerability to consequences.

## 4. Passenger Safety in Indigenous Communities

### 4.1. Indigenous Passengers

Where Indigenous Australians have died in a fatal car crash, the most common cause of occupant injury resulted from a non-collision crash (26%), such as the vehicle running off the road or rolling, with the next most common cause being colliding with another object (23%). Of those Indigenous Australians fatally injured, the number of driver and passenger fatalities was comparatively similar. This contrasts to non-Indigenous Australian fatalities, where there were almost three times as many driver fatalities compared to passenger fatalities. The discrepancy between rates of serious injury for Indigenous and non-Indigenous passengers is ubiquitous over all ages, but peaks for younger drivers (refer to Figure 1) [13]. The number of serious injury incidences for Indigenous vehicle passengers and the discrepancy in such incidences between Indigenous and non-Indigenous Australians are proportional to regional remoteness, with rates highest in very remote regions, and decreasing through remote, outer-regional, inner-regional, and major cities.

**Figure 1 ijerph-18-02446-f001:**
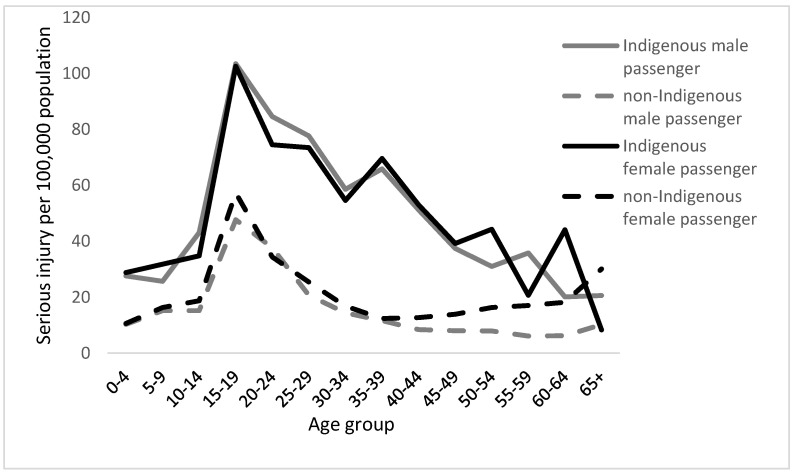
Age-specific rates of serious injury to male and female Indigenous and non-Indigenous passengers from 2005–2006 to 2009–2010 in Western Australia. Adapted from [28].

### 4.2. Passenger Behaviour—Seatbelt Use

Seatbelt use has been mandatory in all states and territories in Australia since 1982 for all vehicles, including trucks and buses [29]. Among all Australians, seatbelt usage rates are among the highest in the world [30]. However, in the Northern Territory, which has the highest rate of Indigenous road fatalities, crash data indicate that 68% of these road fatalities are as a consequence of failing to use a seatbelt. This was the highest behavioural risk factor associated with Indigenous road deaths, slightly higher than driving over the legal BAC limit [27] (see Figure 2).

The authors of [9] identified lack of seatbelt use as one of the common themes reported by Indigenous people who were injured in transport incidences. The reasons that participants reported for not using protective equipment such as seatbelts were multifaceted, but frequently involved the theme of “not going far”.

“*… If we’re not going far, we don’t worry…*”(p. 6)

“*… We didn’t have our belts on either—seemed silly to put them on for a couple of streets…*”(p. 7)

Related to this theme, Indigenous crash victims also reported travelling unrestrained in the back of utility vehicles (“utes” in Australian parlance, also known elsewhere as “pickup trucks”).

“*A whole crew of us were piled into the work ute heading back… I had hit some loose gravel… A couple of boys in the tray were thrown out. One flew through the air like a rag doll*.”(p. 7)

“*… Our mate pulled up in his ute and we all jumped in the tray-back and he rolled it on the next corner*.”(p. 7)

**Figure 2 ijerph-18-02446-f002:**
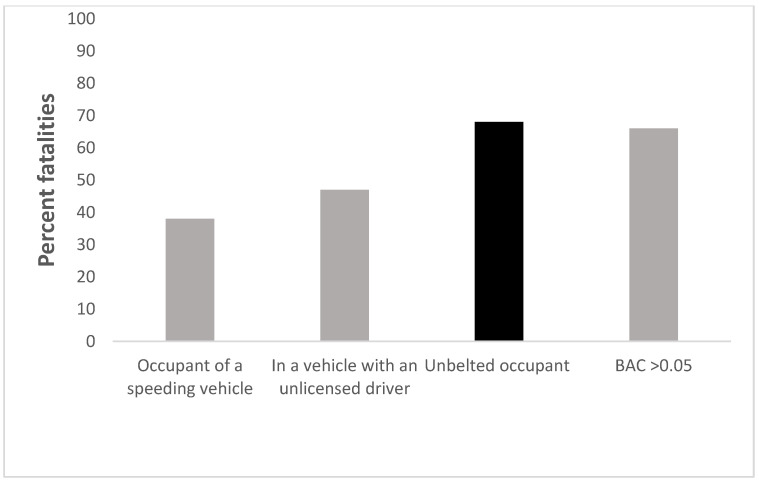
Percentage of road fatalities involving Indigenous peoples in the Northern Territory 1996–1999. Adapted from [31].

### 4.3. Passenger Behaviour—Child Restraints

Australian law requires that children under 4 years of age are restrained in the back seat of the vehicle. Additionally, babies must be in a rear-facing “baby capsule” and, once infants are too big for a baby capsule (at around 6 months of age), they transition to a car seat. It is recommended that infants remain rearward facing in the car seat until approximately 12 months old, before transitioning to forward facing, and then to a booster seat [32]. Children generally grow out of their booster seats around ages 7–10 years. Children over the age of 4 can only be seated in the front seat (in an appropriate child restraint) if the back seat is already occupied by younger children. The driver of a vehicle is responsible for the proper restraint of all passengers under the age of 16 [32]. Data from the Northern Territory show that, of the 50 Aboriginal children killed or seriously injured in car accidents from 2005–2015, 92% were not properly restrained in a baby capsule, car seat, or booster seat [33]. Indigenous children under the age of 4 years are four times more likely than their non-Indigenous counterparts to experience serious injury in a traffic-related incident [1].

The authors of [34] were some of the few researchers to have explored some of the reasons surrounding the low rate of usage of child restraints in Australian Indigenous populations. Their research employed survey and qualitative methods to explore parental use, knowledge, and attitudes of childhood vehicle restraints. The authors demonstrated that Indigenous and non-Indigenous parents were equally likely to report using age-appropriate car seats. For all parents at the childcare centre where the study was conducted, knowledge of the safety qualities of child seats and restraints predicted their use, and younger children (2–3 years) were less likely to be restrained correctly than older children (4–5 years). However, this part of the study did not distinguish between Indigenous and non-indigenous families; thus, Indigenous-specific responses cannot be determined. The focus groups identified that some of the factors impacting the use of child-appropriate restraints included knowledge around children’s seatbelt laws, as well as peer influences. This value of this study was that it explored these issues using a mixed methods approach (i.e., both qualitative and quantitative methods); however, it was limited by its inability to differentiate between, and compare, Indigenous and non-Indigenous responses.

Compliance with child restraint requirements is hampered by knowledge of the appropriate restraints for the different age ranges. In Australia, children must be in restraints that are appropriate to the size and weight of the child. However, when questioned, 50% of Indigenous families reported the incorrect restraint for children 3–4 years, and 76% of Indigenous parents/carers identified the incorrect restraint for 5–8 year old children [27]. It is likely that this is a consequence of limited dissemination of important factual information regarding the use of childhood restraints in remote and very remote Indigenous communities. While there have been some public education campaigns aimed at highlighting the importance of appropriate child restraints (we return to this theme in Section 4 of this paper), such interventions have often been highly localised (e.g., Daruk Aboriginal Health Service in greater Western Sydney), sporadic, poorly coordinated and funded, and not subject to scientific evaluation.

There is an interesting tension in these findings; although government reports indicate that failure to use in-car restraints is associated with a high number of traffic offences, as well as increased serious injury and fatality for Indigenous peoples, qualitative studies demonstrate that the reported use of restraints by Indigenous peoples is high. For example, [35] indicated that 98% of Indigenous people interviewed reported “always” wearing a seatbelt as a driver or passenger. This relative discrepancy between a high rate of individual reported use versus crash data reports mirrors reported versus actual compliance with seatbelt laws for the at-risk demographic of young Indigenous drivers (16–24 years) who reported 80% compliance with seatbelt laws. The authors of [36] similarly described that 77% of their Indigenous parents and carers reported compliance with child restraint laws (compared with 85% of non-Indigenous parents/carers), yet crash data suggest that over 60% of Indigenous passenger fatalities were associated with not using a seatbelt or appropriate restraint. The possible reasons for this discrepancy are numerous, but point to the likelihood that noncompliance is unlikely to be a consequence of wilful neglect or misunderstanding, but rather a consequence of the social and systemic barriers that limit equitable access to services and information in Indigenous communities. For example, because of limited access to safe vehicles, child restraints are often removed to make way for additional people in a car, children often travel in other people’s cars, and child restraints are three times more expensive to purchase in remote communities than in major cities [33].

The social and attitudinal context of seatbelt use is, therefore, important in understanding this complex issue. In many cases, Indigenous parents do not trust seat restraints and/or believe that the child’s comfort and happiness override any potential safety issue. Here, a typical scenario would be that a fractious child is taken out of a restraint to be comforted, or a parent believes that they are better able to keep their child safe [27]. Moreover, a common methodological consideration in qualitative studies with Indigenous populations involves respondents’ notions of shame and uncertainty, which may influence likely responses, thus making it difficult to effectively assess research outcomes. For example, discrepancies between reported restraint use (seatbelt or child restraint) and actual restraint use may be associated with social desirability bias or acquiescence bias of Indigenous respondents, such that they avoided reporting non-seatbelt or restraint use when interviewed alongside non-Indigenous colleagues or when interviewed by non-Indigenous researchers, in an effort to avoid feeling shame and embarrassment. Similar methodological concerns associated with cross-cultural response biases have been reported elsewhere in various other research and applied domains concerning Indigenous populations [37,38,39].

### 4.4. Passenger Behaviour—Overcrowding

In a qualitative study of road safety behaviours and attitudes, [35] reported that rates of seatbelt use in rear seats was low for Indigenous populations compared with non-Indigenous populations. They suggested that this is associated with overcrowding in cars, resulting in insufficient restraints being available to service all passengers.

Overcrowding occurs partly because there are many communities where few people have their licence and there are few roadworthy vehicles. The consequence of this is that there is a considerable burden of responsibility for those people with a licence and who have access to a registered vehicle. This is particularly important given the cultural Indigenous expectation of reciprocity and sharing of resources. Like many collectivist cultures, those who live in remote Indigenous communities typically have at their core a philosophy where community ownership is primary, which contrasts with Western individualist notions of individual behaviour and ownership [29,40]. Here, the individual is then beholden to the community in their goods and deeds. Thus, the car owner/driver becomes a community resource with a strong obligation to the community. With such a limited resource, overuse in the form of vehicle overcrowding is inevitable.

## 5. Critical Evaluations of Previous Strategies

A number of general road safety interventions have targeted Indigenous communities [41], but few have specifically targeted seatbelt use, and none have specifically targeted overcrowding in vehicles. Moreover, of those interventions that have been implemented, very few follow-up evaluations were conducted to determine overall efficacy. This limitation is consistent with the broader road safety space where government-initiated interventions target a range of behaviours from drink-driving to fatigue, but initiatives are frequently developed without guiding theory or evidence, without a thorough understanding of the target audience, and without being scientifically evaluated [42,43]. This is an area that government and community must prioritise moving forward, as the only way to understand the effectiveness of road safety strategies is by developing and implementing targeted, evidence-based interventions that are followed up with critical evaluation. This will ensure that appropriate funding is allocated in the most productive and cost-effective manner to strategies that truly work.

### 5.1. Buckle Up Safely

The Buckle Up Safely (Shoalhaven) program was spearheaded by The George Institute for Global Health in 2010 [34] and has at its core the precaution adoption process model [44]. This was a program targeted toward Indigenous communities, but derived from a bigger program designed to increase the appropriate use of child seats and restraints in children 0–5 years. In the program, childcare staff and parents were provided with information illustrating the importance and use of appropriate child restraints. We found one published evaluation of this program in the Indigenous driving context, where [34] implemented and evaluated the Buckle Up Safely program in a regional area of the state of NSW in Australia that has a high number of Aboriginal families. The program involved educating childcare staff on the safety benefits of the appropriate use of child restraints and explaining how that information could be disseminated to parents and children via the childcare centre. Parents were given information-packs containing safety-relevant resources and education sessions. A strength of this program was the availability of cost-subsidised child seats and access to authorised fitting stations. This was an important component of the initiative as it obviates the financial barriers that otherwise contribute to child-restraint non-compliance.

The evaluation found that this initiative did not significantly increase children’s restraint use in this population; however, self-reported child restraint use was already reasonably high at 87% reported compliance. The study did, however, highlight additional contributing issues that reflect recurring themes in the literature. Almost 30% of families were not able to take advantage of the subsidised restraints because of financial hardship. The outcomes were also subject to the vagrancies of attempting to conduct community research, with the results impacted by factors such as staff turnover, time-poor parents, and the likelihood of social desirability in reporting. The findings were also difficult to attribute to Indigenous populations as the authors were unable to target the Indigenous community in this case; rather, they relied on the relatively high proportion of Indigenous children attending the childcare centres (~31%). Nevertheless, the authors refer to the program as “showing promise”, as families had a higher likelihood of better restraint use as a result of participating in the program, and the program should be commended on being firmly rooted in evidence-based theory.

### 5.2. Bring the Mob Home Safely

Bring the mob home safely was a safe systems approach that was not limited to passenger safety, but included safe cars, speeds, roads, and footpaths, as well as pedestrian and bicycle safety. This intervention was message-based and ubiquitous across communication platforms, including print, radio, merchandise, television, and social media. The strength of this intervention was to build culturally appropriate communication strategies targeting Aboriginal obligations and responsibility to community. However, as far as we are aware, no evaluation of saturation, impact, or outcomes has been conducted on this initiative.

### 5.3. Other, Smaller Strategies

Other, smaller strategies targeting seatbelt use include Yarnbusters: No Gammin DVD, a suite of short videos to identify the importance of restraints, overcrowding, and safe vehicles in the context of purchasing a vehicle [45]. Buckle Them Up—Aboriginal Seat Belt Campaign and Buckle Up Borroloola were programs targeting specific Aboriginal regions and communities (the Pitjantjatjara in South Australia and Borroloola in the Northern Territory communities respectively) [46,47]. Buckle Them Up involved the provision of information resource packs consisting of videos, pamphlets, and training resources, and it had a strong message about the impact of death and serious injury in Aboriginal communities. The resources were delivered in both English and Pitjantjatjara languages and developed and managed by Pitjantjara community members. The community involvement in the development and implementation of this program was one of its strengths, giving the community ownership of the issue, in a way that incorporates their community ways of knowing, being, and doing. Buckle Up Borroloola was a joint initiative with the Northern Territory Accidents Compensation Commission (MACC) and KidSafe Northern Territory to provide affordable child seat restraints and appropriate fitting to families in the Borroloola region. This was similar to the Child Restraint Program for Aboriginal Communities [48], which was an initiative with the Centre for Road Safety and the Community Transport Association, which similarly provided child restraints and child restraint fittings to Aboriginal communities across New South Wales.

However, once again, there is no evidence of these programs continuing, and no evaluation of saturation, impact, or outcomes. Their strength is that, in general, the programs have been driven by community and, thus, culturally sensitive and targeted to each community’s unique needs. However, the limitations are that most of these programs (with the exception of Buckle Up Safely) are not framed within a sound evidence base, and that there is limited evidence for the efficacy of the programs in eliciting behaviour change. This highlights the tension between designing and conducting evidence-based interventions, which can be time-consuming, complex and rigid in methodology, compared with community-driven research which is respectful, fluid, interpretive, and iterative.

## 6. Directions for Future Research

In a review of road trauma involving Indigenous peoples, [49] indicated that we have very little understanding of the underlying causes of risky driving behaviour in Indigenous communities. Interventions have been largely ineffective, and it is unclear whether this is a consequence of lack of opportunity, lack of understanding, or messaging simply failing to target those people most at risk. This review is over 10 years old, and little has changed in the intervening years. Advancements in health and safety interventions in Indigenous road trauma must involve messaging that is derived from a strong basis in behavioural theory and lived experience of behaviour change. In other words, Indigenous communities must be given the opportunity to not only hear effective road-safety messages, but also experience them, and, most importantly, they must be involved in generating culturally appropriate road safety messaging and education initiatives.

### 6.1. Theory-Driven Messaging

Broadly, the aim of road safety public education messages is to encourage safer road user behaviours [50]. To this end, Australia has typically adopted the use of fear-based messages which depict death or physical injury as the outcome of the risky behaviour [50,51]. Fear-based advertisements are based on the premise that frightening the target audience will be an appropriate motivator for behaviour change; however, their effectiveness in reducing crash rates is inconclusive and the public is growing numb to them [47]. Indeed, research has found that other key theoretical constructs have a greater impact on message persuasiveness, such as a focus on the key motivations for engaging in the risky behaviour, appropriate modelling of the desired behaviour, and the inclusion of practical strategies to reduce the risky behaviour [52,53,54]. While messages based on theory are more effective than those that are not [55], messages are still being developed without guiding theory [56].

Decisions regarding road safety public education messages are often the job of marketing and advertising personnel, many of whom are unaware of behaviour change theories from disciplines such as psychology [57]. A relatively new framework, The Step Approach to Message Design and Testing (SatMDT) [54], was specifically devised to guide the development and evaluation of road safety public education messages and is based on the underlying principles derived from a range of social psychological theories of decision-making and attitude–behaviour relationships such as the Theory of Planned Behaviour [58], the Extended Parallel Process Model [51], the Elaboration Likelihood Model [59], and Social Learning Theory [60]. These principles support the view that advertising can directly motivate changes in attitudes, intentions, and behaviour when firmly embedded in psychological theory [42,61]. In addition to these sound theoretical underpinnings, they emphasise the importance of gaining a thorough understanding of the target population, piloting the messages with the target population, and applying rigorous scientific methods to evaluate of the messages’ effectiveness. The SatMDT has successfully been applied to the development and evaluation of road safety messages in various contexts, such as smartphone use among young drivers [62,63] and the use of child restraints [61]. The application of the SatMDT to the Indigenous road safety context would improve the likelihood that messages are effective in reducing risky road behaviours (e.g., increasing the use of seatbelts) among Indigenous road users.

### 6.2. The Nexus Between Western Approaches to Learning and Indigenous Ways of Knowing, Being, and Doing

Indigenous drivers and passengers in Australia are overrepresented in crash statistics compared to non-Indigenous Australians. One potential explanation for this gap could be that Western ways of road safety messaging are neither applicable nor relevant to Indigenous Australians. Attempting to implement non-Indigenous strategies, methods, and interventions in Indigenous communities, which hold different cultural values and norms, results in misunderstanding, miscommunication, and mistrust. Elsewhere, educational programs based on Western perspectives have been criticised for failing to consider culturally specific values, goals, domains, languages, and learning styles [64,65,66], and there is a push for interventions that are responsive to the culture and context of the local community [67]. While traditional Indigenous learning relies largely on visual observation, imitation, and oral transmission of knowledge, Western educational styles are typically more instructional and more directive, and they rely more on the transmission of written information [68]. This disjuncture is evident in the area of driver licencing attainment, where Indigenous and non-Indigenous pre-learners are expected to learn road safety rules by studying a written road safety manual, such as the NSW Road Users’ Handbook, and are subsequently tested on their knowledge via a written exam. This method of learning and testing is based on an assumption that the pre-learner has the fundamental literacy skills required to read and comprehend the material provided, which serves as a cultural and linguistic barrier to those Indigenous peoples who may not possess the necessary literacy skills.

Effective advertising messages need to be specifically targeted to the end user and developed with a deep understanding of the target group’s key motivations for engaging in risky behaviours [61]. Road safety messages for Indigenous communities, therefore, need to be more nuanced and targeted, driven by a synthesis between Indigenous culture, belief, and world view and fundamental psychological theory. It is time that evidence-based messaging is firmly rooted in Indigenous cultural knowledge and developed with an understanding of the beliefs, motivations, and barriers that underpin risky driving behaviour. The limited research that has been conducted on Indigenous messaging has found that there are very few aspects of mainstream road safety messages that are relevant to Indigenous communities. Specifically, the factors that may motivate behaviour change for non-Indigenous Australians are unlikely to motivate behaviour change for Indigenous peoples in remote and very remote communities as they are misaligned with Indigenous cultural norms and motivations [42]. This suggests that this is an area ripe for exploration. Identifying the key motivations for Indigenous peoples’ risky driving behaviours is vital if effective road safety messaging is to be developed and behaviour change is to occur.

The aim going forward is to synthesise psychological theory and Indigenous ways of knowing, being, and doing. In particular, this can be achieved by considering combining theory-based messaging with experiential learning that is more immersive, authentic, and practical, driven by Indigenous cultural epistemologies, and responsive to the context of the local community.

### 6.3. A Novel Solution: Immersive Technology as a Learning Tool

The possibilities afforded by immersive technologies such as virtual reality (VR) and simulation as an authentic learning tool are currently being explored in the University of Newcastle’s Applied Psychology laboratory and elsewhere [69]. The efficacy of virtual environments can be explored for two purposes. The first is allowing drivers to gain driving skills in a safe environment where the operator can be challenged by dangerous driving environments that might otherwise only be encountered under emergency conditions. However, it is the second (i.e., the opportunity to learn the rules and regulations of driving) where we feel VR holds the most promise. The value of simulation as a pedagogical tool is well known [70], with the strength of VR lying in its interactive nature with the operator having a feeling of being present, leading to a real-life, immersive experience of the learning event [71]. This has allowed operators to explore complex situations and environments in novel ways, embedded in relevant context, and it synchronises nicely with traditional Indigenous learning strategies. The value of this for passenger safety is twofold: an immersive, culturally safe learning environment allows novice drivers to more effectively learn the necessary rules and legislation around keeping their passengers safe. Similarly, a safe, but realistic environment allows an authentic experience of the potential consequences of failing to use a seatbelt and/or overcrowding a vehicle.

Too frequently, Western expectations of knowledge acquisition fail to synchronise with Indigenous epistemologies. “For Aboriginal people, learning is a journey and it is not so much about the self; it is about the journey you take when you are sharing those experiences” (quoted from a participant in [72] (p.136), where relational epistemology is central to the learning journey [73]. In Australia, learning road rules and legislative expectations are core steps involved in gaining a driver’s licence. Obtaining and maintaining a driver’s licence and adopting safe driving habits such as seatbelt use require adherence to the national and federal road rules, and access to such material is typically through written pamphlets, booklets, and websites. This provides a considerable barrier to individuals who may have inadequate access to resources and who have limited English language proficiency and associated literacy skills. There is a sociocultural obligation to embed the rules and regulations required to operate a vehicle safely, within an epistemological framework that enables better access and a better learning environment for Indigenous peoples.

The authors of [69] considered the efficacy of driving simulators in road safety education in Indigenous communities. In their scoping review, they found no education programs using simulated environments targeting Indigenous communities. This is extraordinary given the value that VR learning environments could bring to Indigenous road safety education; the immersive, contextualised nature of VR technology has a much closer alignment to traditional ways of knowing, being, and doing than traditional methods of learning road rules and legislation, which is typically text-based. Such technology also enables greater access for Indigenous peoples who may be unfamiliar with written English and/or have literacy limitations.

The current mechanisms via which people access and learn road rules and safe practices do not contextualise the learning experience, and they are themselves derived from a Western framework. We need to strive for cultural inclusivity when it comes to understanding road rules and legislation, as well as road safety messaging. VR technology has become a financially feasible option, with self-contained head-mounted displays retailing for less than $200, making it possible for schools, colleges, libraries, and youth centres to make them available as effective learning resources.

## 7. Conclusions

Despite evidence spanning 20 years indicating that Australian Indigenous passengers in cars are at greater risk than non-Indigenous passengers, it remains a complex yet unresolved issue, connected to issues of culture, socioeconomic status, and systemic barriers associated with equitable access to necessary road safety resources. Of all transport fatalities and serious injuries, Australian Indigenous and non-Indigenous people are equally represented, such that around 50% of transport fatalities and serious injuries for Indigenous and non-Indigenous people are from being the occupant of a car. The difference is in the proportion of driver versus passenger trauma, such that Indigenous deaths and serious injuries are twice as likely for passengers versus drivers, while non-Indigenous drivers are more likely to suffer fatalities and injuries compared to passengers. Most recently, [35] (p127) suggested that “*targeted community-led programmes around restraint use are still urgently needed, particularly in rural and remote areas. It is likely that overloading and lack of restraint use are major contributors to the over-representation of Aboriginal and Torres Strait Islander people in road deaths, and this requires urgent, targeted intervention*”. Broadly, the development and the evaluation of effective road safety messages based on theory and designed with a thorough and nuanced understanding of Indigenous people are imperative to increase restraint usage among the Indigenous community. More specifically, the development of innovative and technology-driven learning affords us vast and unparalleled opportunities for creative and culturally responsive environments for Indigenous Australians to learn safe driving rules and practices in a meaningful and authentic way.

## Data Availability

Not applicable.

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
