# Peer review of "Keeping Safe on Australian Roads: Overview of Key Determinants of Risky Driving, Passenger Injury, and Fatalities for Indigenous Populations"

_ijerph, 2021, doi:10.3390/ijerph18052446_

Round 1
Reviewer 1 Report
This paper focuses on the traffic accidents of indigenous people in Australia and summarises the key factors of passenger injury and fatalities. Previous strategies for improving the traffic safety for indigenous people are also evaluated and future research directions are pointed out. The paper is well written, and the structure is easy to follow. The reviewer has several comments as follows:
- It may be easier for the reader to understand if the risky driving behaviours (Section 4) are first summarised and the reasons for these risky driving behaviours (Section 3) are then explained.
- The reviewer suggests that the authors should provide more general summarisation of the limitations of existing studies of risky driving behaviours, their underlying causes and intervention strategies, and thus the future directions corresponding to these limitations.
- It seems that Section 7 provides another future research direction. What is the reason it is in a separated section.
- The format of references needs to be unified as it is in different format for [12][20], etc.
Author Response
- This paper focuses on the traffic accidents of indigenous people in Australia and summarises the key factors of passenger injury and fatalities. Previous strategies for improving the traffic safety for indigenous people are also evaluated and future research directions are pointed out. The paper is well written, and the structure is easy to follow.
Response: We thank the reviewer for their positive evaluation of the manuscript.
- It may be easier for the reader to understand if the risky driving behaviours (Section 4) are first summarised and the reasons for these risky driving behaviours (Section 3) are then explained.
Response: Thank you for this suggestion. We have moved some of the text around such that the summary of risky driving is at the start of Section 3, followed by a more thorough discussion of the systemic barriers. We feel however that it is important that the associated systemic barriers should be acknowledged up front and early in order to contextualise the unsafe behaviour, so as not to perpetuate a deficit narrative around the behaviour.
- The reviewer suggests that the authors should provide more general summarisation of the limitations of existing studies of risky driving behaviours, their underlying causes and intervention strategies, and thus the future directions corresponding to these limitations.
Response: A general summary is provided before describing the interventions
A number of general road safety interventions have targeted Indigenous communities [37], but few have specifically targeted seatbelt use, and none have specifically targeted overcrowding in vehicles. Moreover, of those interventions that have been implemented, very few follow-up evaluations were conducted to determine overall efficacy. This limita-tion is consistent with the broader road safety space where government-initiated interven-tions target a range of behaviours from drink driving to fatigue, but initiatives are fre-quently developed without guiding theory or evidence, without a thorough understanding of the target audience, and without being scientifically evaluated [38,39]. This is an area that government and community must prioritise moving forward, as the only way to un-derstand the effectiveness of road safety strategies is by developing and implementing targeted, evidence-based interventions that are followed up with critical evaluation. This will ensure that appropriate funding is allocated in the most productive and cost-effective manner to strategies that truly work.
And we have further summarised the issues as a pointer to future research to be discussed in the subsequent section:
However, once again, there is no evidence of these programs continuing, and no evaluation of saturation, impact or outcomes. Their strength is that in general the pro-grams have been driven by community and thus culturally sensitive and targeted to each community’s unique needs. However, the limitations are that most of these programs (with the exception of Buckle-up Safely) are not framed within a sound evidence-base, and that there is limited evidence for the efficacy of the programs in eliciting behaviour change. This highlights the tension between designing and conducting evidence-based interven-tions, which can be time consuming, complex and rigid in methodology, compared with community-driven research which is respectful, fluid, interpretive and iterative.
- Directions for Future Research
In a review of road trauma involving Indigenous peoples, [41] indicated that we have very little understanding of the underlying causes of risky driving behaviour in Indige-nous communities. Interventions have been largely ineffective, and it is unclear whether this is a consequence of lack of opportunity, lack of understanding, or messaging simply failing to target those people most at risk. Clapham’s review is over 10 years old, and little has changed in the intervening years. Advancements in health and safety interventions in Indigenous road trauma must involve messaging that is derived from a strong basis in behavioural theory and lived-experience of behaviour change. In other words, Indigenous communities must be given the opportunity to not only hear effective road-safety messag-es, but also experience them, and most importantly, must be involved in generating cul-turally appropriate road safety messaging and education initiatives.
- It seems that Section 7 provides another future research direction. What is the reason it is in a separated section.
Response: Thank you for this suggestion, we have amended Section 7 to be Section 6.3 and thus falling under ‘future research’
- The format of references needs to be unified as it is in different format for [12][20], etc.
Response: The referencing format has been amended to be consistent with the template requirements.
Reviewer 2 Report
This was a very interesting paper and is worthy of publication - as the authors say, it tackles an issue not frequently discussed by academic sources.
I would like to see clearer parameters being stressed at (77-84)in terms of the focus of the paper and a clearer indication of what 'risky behaviour' is (including also past references and citations to the same).
The paper perhaps would benefit from a tighter analysis of the key factors (with secondary areas being indicated). In other words, at times it felt like the writers jumped around in terms of the focus.
Given the Public Health aspect of the journal, an indication or study as to the consequences/casual effects would be interesting (either within or in a separate publications) - suggestion only.
I would advocate that the authors consider another title for the paper perhaps, which more closely fits their specific parameters - hence defining better the boundaries of this paper (for the reader) - and making this (and the title) inline with the risky behaviour approach and the areas you are focusing on (primarily) within this.
Author Response
Reviewer 2
- This was a very interesting paper and is worthy of publication - as the authors say, it tackles an issue not frequently discussed by academic sources.
Response: We thank the reviewer for their positive evaluation of the manuscript.
- I would like to see clearer parameters being stressed at (77-84) in terms of the focus of the paper and a clearer indication of what 'risky behaviour' is (including also past references and citations to the same).
Response: We thank the reviewer for this suggestion and we have amended for clarity (see paragraph below):
‘Consistent with the theme of this Special Edition, the primary focus of this review paper will be understanding risky driving behaviours that impact passenger safety. Specifically, this paper will focus on the use of seatbelts, child restraints, and over crowded cars, given these behaviours are known contributing factors to death and serious injury among Indigenous Peoples [3]. Importantly, we recognise that to clearly understand the determinants of these risky driving behaviours, our review needs to be embedded within the broader context of risky driver behaviour in Indigenous populations. As such, this paper will consider the general context of risky driving practices in Australian Indigenous communities from a road-safety, socio-cultural, and population perspective.’
Please note that the term ‘risky driving behaviour’ is defined on lines 67-68 as follows:
‘Risky driving behaviour is driving behaviour that increases the potential for harm or serious injury [e.g., 9, 10, 11]’.
In addition, and in accordance with Reviewer 1’s feedback, we have amended section 3 for clarity, and Section 2 also outlines these issues.
- The paper perhaps would benefit from a tighter analysis of the key factors (with secondary areas being indicated). In other words, at times it felt like the writers jumped around in terms of the focus.
Response: We thank the Reviewer for this suggestion. In addressing the other issues discussed we feel that this has resulted in a tighter narrative. We have also added additional text as required to contextualise the secondary areas.
- Given the Public Health aspect of the journal, an indication or study as to the consequences/casual effects would be interesting (either within or in a separate publications) - suggestion only.
Response: We thank the reviewer for this suggestion, which we agree would make an interesting future study/review paper.
- I would advocate that the authors consider another title for the paper perhaps, which more closely fits their specific parameters - hence defining better the boundaries of this paper (for the reader) - and making this (and the title) inline with the risky behaviour approach and the areas you are focusing on (primarily) within this.
Response: We thank the reviewer for this section, we feel that the title captures well the content and focus of the paper, however as requested, we have added “risky driving” to the title to become: Keeping safe on Australian roads: Overview of key determinants of risky driving, passenger injury and fatalities for Indigenous populations